# Short-term activation of the Jun N-terminal kinase pathway in apoptosis-deficient cells of *Drosophila* induces tumorigenesis

Noelia Pinal[1], María Martín[1], Izarne Medina[1] & Ginés Morata[1]

In *Drosophila*, the JNK pathway eliminates by apoptosis aberrant cells that appear in development. It also performs other functions associated with cell proliferation, but analysis of the latter is hindered by the pro-apoptotic activity. We report the response of apoptosis-deficient cells to transient activation of JNK and show that it causes persistent JNK function during the rest of the development. As a consequence, there is continuous activity of the downstream pathways JAK/STAT, Wg and Dpp, which results in tumour overgrowths. We also show that the oncogenic potential of the Ras-MAPK pathway resides largely on its ability to suppress apoptosis. It has been proposed that a hallmark of tumour cells is that they can evade apoptosis. In reverse, we propose that, in *Drosophila*, apoptosis-deficient cells become tumorigenic due to their property of acquiring persistent JNK activity after stress events that are inconsequential in tissues in which cells are open to apoptosis.

[1] Centro de Biología Molecular, CSIC-UAM, Madrid 28049, Spain. Correspondence and requests for materials should be addressed to G.M. (email: gmorata@cbm.csic.es)

Apoptosis, i.e. programmed cell death, is a physiological process by which cells trigger their own destruction. Although there is developmentally programmed apoptosis, i.e. the removal of the tail of tadpoles or the articulations of *Drosophila* legs[1,2], a major function of apoptosis is the elimination of aberrant cells that may compromise the viability or the fitness of the organism[3–5]. Apoptosis also functions to remove, through a process known as cell competition[3,6,7], oncogenic cells that may arise during development[4,8].

The study of apoptosis in *Drosophila* is facilitated by the simplicity of the genetic program, the availability of mutants affecting apoptosis and of methods to manipulate gene activity. Within *Drosophila*, the wing imaginal disc (the precursor cells of the adult wing and thorax) is a convenient experimental system, for while it normally does not contain cells in apoptosis, it exhibits a strong apoptotic response to ionising radiation or other forms of stress[9]. A simplified version of the apoptosis program is shown in Fig. 1a.

A prime event in the apoptotic response in vertebrates and *Drosophila*[10–12] is the activation of the Jun N-terminal kinase (JNK) signalling pathway, a member of the MAPKs family involved in various cellular processes[13]. In the *Drosophila* wing disc, JNK is not normally active—except in a small proximal region (Fig. 1b, c)—but after irradiation, there is overall induction of JNK and subsequent apoptosis[10,11] (Fig. 1d, e), aimed to remove cells damaged by the irradiation. The apoptotic role of JNK also mediates the elimination of oncogenic cells by cell competition, thus revealing its anti-tumour role[4,14].

In addition to its apoptosis-inducing function, JNK is known to play non-apoptotic roles connected with cell proliferation[15–17]. For example, sustained expression of JNK is associated with cancer in vertebrates[18,19], and in *Drosophila* we have shown that forcing JNK activity causes overgrowths in apoptosis-defective wing discs[17]. Thus, both in flies and in vertebrates, JNK may function as anti-tumorigenic or pro-tumorigenic, depending on the context.

Another factor associated with tumorigenesis in vertebrates is the inappropriate expression of the Ras-MAP kinase pathway[20,21]. In *Drosophila*, Ras overexpression can also cause massive overgrowths through interactions with mutations at tumour suppressing genes[22,23]. Significantly, the development of those overgrowths requires JNK activity[3,23]. One noteworthy feature of the overexpression of Ras-MAPK in *Drosophila* is that it makes cells refractory to apoptosis[24,25], and a similar observation has been made in mammalian cells[26].

Our results indicate that cells refractory to apoptosis, they may lack the apoptosis machinery or overexpress the Ras pathway, become tumorigenic due to their property of acquiring sustained JNK activity after its transient activation due to stress or other events.

## Results

**JNK sustained activity in apoptosis-defective cells.** In our experiments, we generated entire discs or disc domains that cannot initiate apoptosis. They may be mutant for the caspase Dronc, or may lack activity of pro-apoptotic genes (see Fig. 1 and Supplementary Fig. 1 for details), or may contain constitutive expression of the Ras-MAPK pathway[24,25]. Then, we analysed the response to events leading to JNK activation such as X-rays, a pulse of *p53*, or a pulse of JNK itself.

We first analysed JNK response to ionising radiation (3-4000R) using the targets *puckered (puc)* and *metalloprotease 1 (mmp1)* to monitor JNK function. Except for the resident JNK expression in a small proximal region (Fig. 1b, c), any additional activity has to be caused by the experimental intervention. We compared the response of discs in which apoptosis is not prevented (controls) with that of discs in which cells cannot undergo apoptosis.

In irradiated discs open to apoptosis, we observe high *puc* levels 24 h after the irradiation (Fig. 1d), but afterwards those levels decay and eventually disappear by 96 h (Fig. 1f). Cells expressing *puc* also exhibit Dcp1 effector caspase activity indicating that they are in apoptosis. These results confirm previous work and illustrate the JNK role of removing cells damaged by irradiation[10,11].

The JNK response to irradiation of apoptosis-deficient discs is very different.

Although the apoptotic response is suppressed, there is a strong activation of the JNK marker *puc*, especially in the pouch 24 h after X-rays (Fig. 1e). However, unlike the response of *dronc+*, mutant *dronc−* discs maintain high *puc* levels for the rest of the larval development, including pre-pupal stages (Fig. 1g).

We find similar results in discs in which apoptosis is suppressed only in the posterior compartment: 96 h after irradiation, there are high levels of JNK in the posterior but not in the anterior compartment (Supplementary Fig. 1a, b). Further evidence comes from clones homozygous for the *H99* deletion (lacking major pro-apoptotic genes), many of which express JNK markers 96 h after the irradiation, whereas surrounding cells do not (Supplementary Fig. 1c).

We also performed experiments to follow the lineage of the cells that acquire JNK activity after irradiation, both in cells open to apoptosis (*dronc+*) and in apoptosis-defective (*dronc−*) cells.

Irradiation of *dronc+* discs of genotype *puc-Gal4 UAS-Flp UAS-GFPDbox act<stop>lacZ* activates *puc-Gal4* in many of their cells. These cells become labelled with GFPDbox, an unstable form of GFP[27], to avoid an effect of perdurance, and also produce high levels of the recombinase Flippase. In the majority of those cells, Flippase induces recombination in the *act<stop>lacZ* cassette, what labels them and their progeny indelibly with *LacZ*. The *puc*-expressing cells that do not die generate clones marked with *LacZ* expression (see Supplementary Fig. 1 for details). Twenty-four hours after X-rays, we find a large number of cells expressing GFPDbox and the majority of them are also labelled with *lacZ* (Supplementary Fig. 1d). Seventy-two hours after irradiation, we observed a smaller number of large LacZ patches, but none of them expresses GFPDbox (Supplementary Fig. 1e), indicating that JNK has been turned off, consistent with the results above (Fig. 1f). The large size of those patches suggests that the mature disc has been reconstituted by few cells, many of which are descendants from those in which JNK was activated by the irradiation. This is proof that X-rays can activate JNK to sub-lethal levels.

The same lineage analysis after irradiation of *dronc−* discs yields different results. The principal difference is that 72 h after the irradiation, we still observed many groups of cells expressing GFPDbox (Supplementary Fig. 1g), again indicating the persistence of JNK expression in apoptotic defective cells and reinforcing the results above (Fig. 1g). In these discs, the size of the *lacZ* patches is smaller than in the *dronc+* discs, which is expected because the number of cells is greater as there is no cell death after the irradiation.

To explore the possibility that just an initiation event is sufficient to generate persistent JNK activity in apoptosis-defective cells, we carried out two experiments in which we checked JNK expression several days after a short pulse of induction (see Methods for details). In the first experiment, a 16 h pulse of the JNK activator *p53*[28] in the posterior compartment causes persistent JNK activity, visualised 96 h after the pulse (Fig. 1i). In the second experiment, we obtain a similar result after a 16 h pulse of JNK itself (Fig. 1k), achieved by the expression of a form of *hemipterous (hep^CA*, a JNKK[4]), which

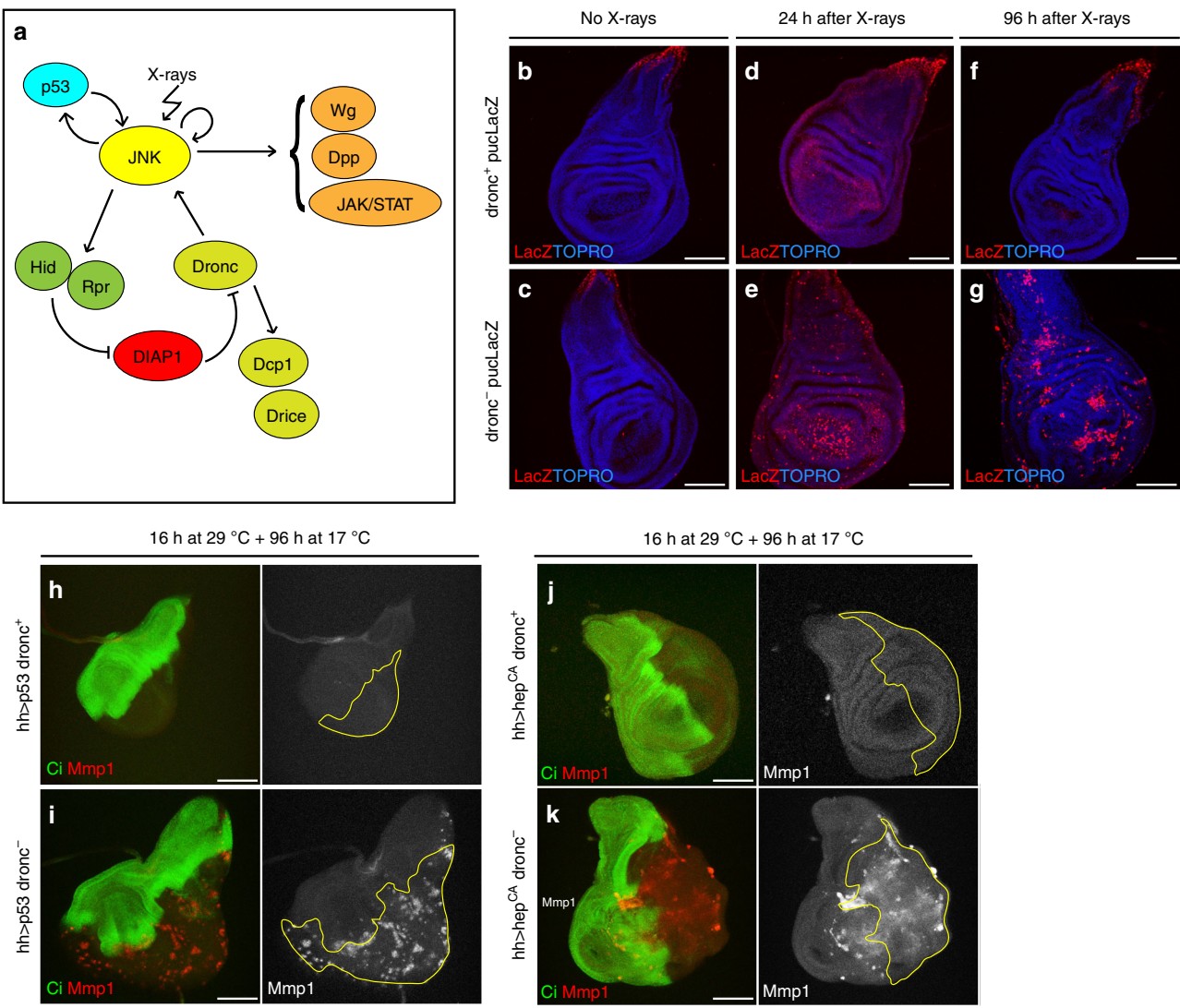

**Fig. 1** JNK persistent activity after irradiation or a short pulse of *p53* and JNK in apoptosis-deficient cells. **a** Simplified representation of the apoptotic pathway in *Drosophila*. JNK/p53 are activated after irradiation and induce the activation of the pro-apoptotic genes that block DIAP1, inducing the activation of the apical caspase Dronc and effector caspases Dcp1 and Drice. Besides, Dronc reinforces JNK activity generating an amplification loop that augments the apoptotic response[26]. As a side event, JNK activates the Dpp, Wg and JAK/STAT pathways. **b-g** Wing discs showing the effects of X-rays at different time points. JNK activity is monitored by the expression of a *LacZ* insert at the JNK target *puckered* (*puc*, red). The blue background reflects TOPRO staining. **b**, **c** *puc* expression in non-irradiated *dronc*+ control ($n = 10$) and null *dronc*− mutant ($n = 17$) discs. Note that the label is restricted in both cases to a small region that corresponds to the most proximal zone of the disc. **d**, **e** Twenty-four hours after irradiation, both *dronc*+ ($n = 22$) and *dronc*− ($n = 18$) discs show high expression levels of *puc*, especially in the region corresponding to the wing pouch. **f**, **g** *puc* expression 96 h after irradiation in *dronc*+ ($n = 28$) and *dronc*− ($n = 31$) discs. Whereas *puc* levels have returned to normal in *dronc*+ (**f**), they remain high in many zones of the *dronc*− discs (**g**). Wing discs of genotype: **h** *UAS-p53/UAS-GFPtubGal80ts;hh-Gal4dronc^{i29}/+* ($n = 10$), **i** *UAS-p53/UAS-GFPtubGal80ts;hh-Gal4dronc^{i29}/dronc^{i29}* ($n = 14$), **j** *UAS-hep^{CA}/UAS-GFPtubGal80ts;hhGal4dronc^{i29}/+* ($n = 10$) and **k** *UAS-hep^{CA}/UAS-GFPtubGal80ts;hh-Gal4dronc^{i29}/dronc^{i24}* ($n = 11$). Discs are stained with *Cubitus interruptus*, Ci (green) to mark the anterior compartment and Mmp1 (red) to monitor JNK activity. The short pulse of *p53* (**h**, **i**) or of *hep^{CA}* (**j**, **k**), at the posterior compartment is administered by temperature shift, detailed in Methods section; 96 h after the change to 17 °C, no activity of JNK can be detected by Mmp1 staining (red) in the posterior compartment (lack of green) in *p53 dronc*+ (**h**) and *hep^{CA} dronc*+ discs (**j**). In contrast, there is JNK activity (red) at the posterior compartment (lack of green) of *p53 dronc*− (**i**) and in *hep^{CA} dronc*− discs (**k**). Scale bars are 100 μm

causes constitutive activity of the pathway. In control discs, in which apoptosis is not prevented, the brief expression of JNK (Supplementary Fig. 1h) rapidly subsides (Fig. 1h, j).

**ROS and *moladietz* are required for JNK persistent activity.**
The preceding experiments establish that events leading to JNK activation trigger a genetic operation that causes its sustained adventitious function. This phenomenon is of little consequence when cells are open to apoptosis because they die shortly after

JNK induction or recover and the JNK pathway is turned off, but becomes prominent when cells are refractory to apoptosis. This persistent activity cannot be due to the apoptotic amplification loop[28] because in the absence of the function of *dronc* or the pro-apoptotic genes *hid/rpr/grim* (*DfH99* deletion), the loop cannot be completed.

We sought to identify the mechanism(s) responsible for the persistence of JNK activity after a transient initiation event. It has been shown that cellular damage to the wing disc induces a burst of reactive oxygen species (ROS), which in turn activate JNK[29].

Furthermore, recent work on wing disc regeneration[30] has identified the gene *moladietz (mol)*, transcriptionally activated by JNK, which encodes a Duox maturation factor necessary for production of ROS, with subsequent induction of JNK. JNK and *mol* establish a positive amplification loop that ensures prolonged JNK function during the regeneration process[30].

Therefore, we checked the involvement of ROS and *mol* on the phenomenon of persistent JNK function observed in our experiments. First, using dihydroethidium (DHE) staining[31] to monitor ROS production, we found high DHE levels in *dronc⁻* discs 96 h after irradiation (Fig. 2b), whereas DHE is not detected in *dronc⁺* discs (Fig. 2a). Second, *mol* activity, visualised with a *mol-LacZ* insert, is also detected in *dronc⁻* discs 96 h after irradiation (Fig. 2e), but not in controls (Fig. 2d).

We checked the functional requirements of ROS and *mol* activity for the sustenance of JNK function. The reduction of ROS levels in the posterior compartment by overexpression of *Superoxide dismutase* (SOD) and *Catalase* (Cat)[29], or the suppression of *mol* function using a molRNAi construct[30], resulted in reduced JNK activity in both cases 96 h after the irradiation (Fig. 2c, f).

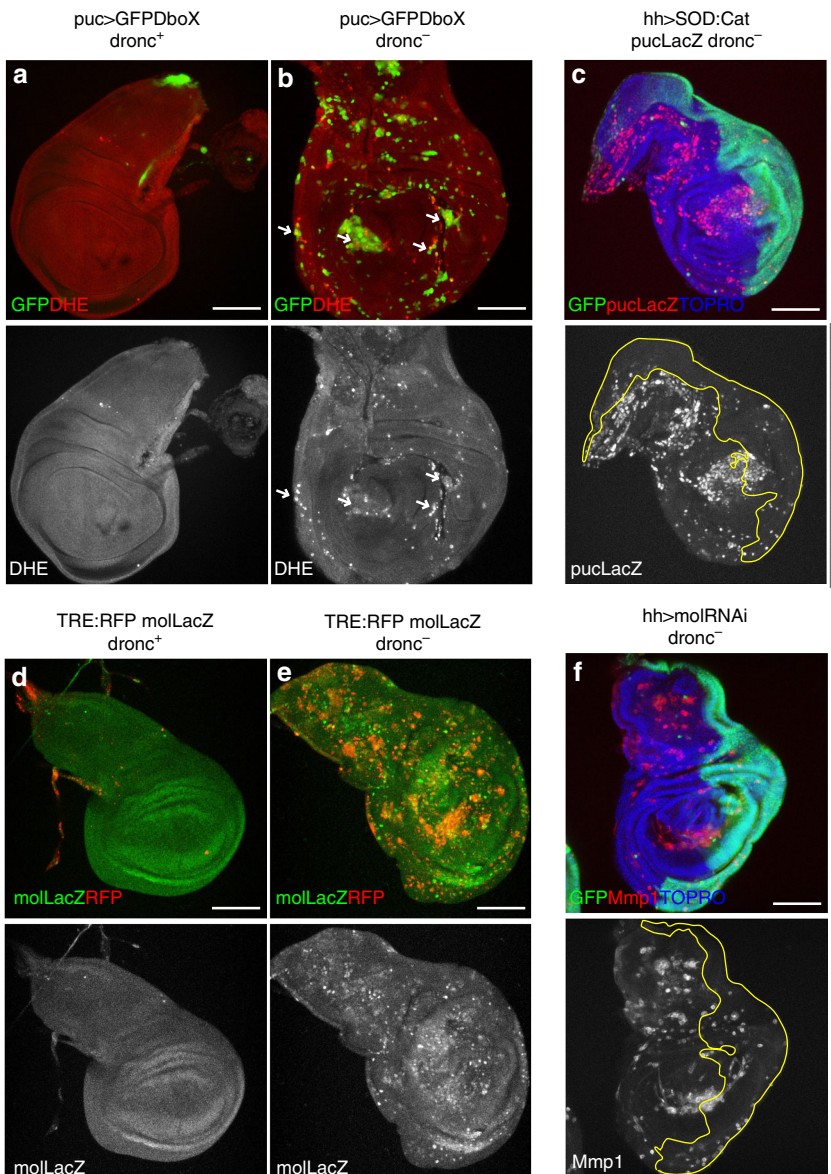

**Fig. 2** ROS and *moladietz* are involved in JNK persistent activity. **a**, **b** Wing discs stained 96 h after irradiation with DHE to detect ROS, **a** *dronc⁺* control (*n* = 10) and **b** *dronc⁻* mutant (*n* = 30). We observe DHE staining in *dronc⁻* mutant disc (**b**, arrows) in cells that present JNK activity, labelled by the expression of *UAS-GFPDbox*, but not in (**a**) *dronc⁺* control. **c** Wing disc of the genotype: *UAS-SOD:UAS-Cat/UAS-GFPtubGal80ᵗˢ;droncⁱ²⁴/droncⁱ²⁹hh-Gal4* 96 h after irradiation (*n* = 14) (after irradiation, larvae were raised for 24 h at 17 °C to allow ROS activation and then transferred to 29 °C for 72 h to block ROS production). We observe a reduction in the activation of JNK (*puc-LacZ*, red) only in the posterior compartment (GFP, green), where *SOD:Cat* are expressed. Wing discs of the genotypes: **d** (*n* = 9) *TRE:RFP/mol-LacZ* and **e** (*n* = 22) *TRE:RFP/mol-LacZ;droncⁱ²⁴/droncⁱ²⁴* 96 h after irradiation where *mol-LacZ* expression (green) is maintained in *dronc⁻* mutants (**e**) in cells that show JNK activity (*TRE:RFP*, red). No RFP or ectopic *mol-LacZ* is observed in (**d**) *dronc⁺* control. **f** Wing disc of the genotype: *UAS-molRNAi/UAS-GFPtubGal80ᵗˢ;droncⁱ²⁴/hh-Gal4droncⁱ²⁹* (*n* = 7) 96 h after irradiation. As in **c**, the mol RNAi was expressed the last 72 h after irradiation. Reduction of the JNK activity (Mmp1, red) is observed in the posterior compartment (green), but not in the anterior. Scale bars are 100 μm

**Sustained JNK induces ectopic expression of target pathways.** The preceding results establish the involvement of ROS and *mol* in the continuity of JNK function after transient activation. Next, we examined whether the persistent JNK function in apoptosis-deficient cells causes continuous mis-expression of pathways downstream JNK like JAK/STAT[23,32], Dpp and Wg[17,33]. Indeed, JAK/STAT, Wg and Dpp are induced after irradiation in territories outside their normal domains, and their expression is associated with the zones of JNK ectopic activity (Fig. 3b, e, g). Moreover, the ectopic expression of JAK/STAT and Wg depends on JNK, for compromising JNK activity with a dominant negative form of Bsk (JNK itself) suppresses their expression (Fig. 3c, h).

**Sustained JNK induces tissue overgrowth.** We also observed a correlation between the continuous expression of JNK and an increase of cell proliferation, the latter monitored by EdU incorporation, which results in the formation of overgrowths (Fig. 4a, b, d). We also find that the development of the

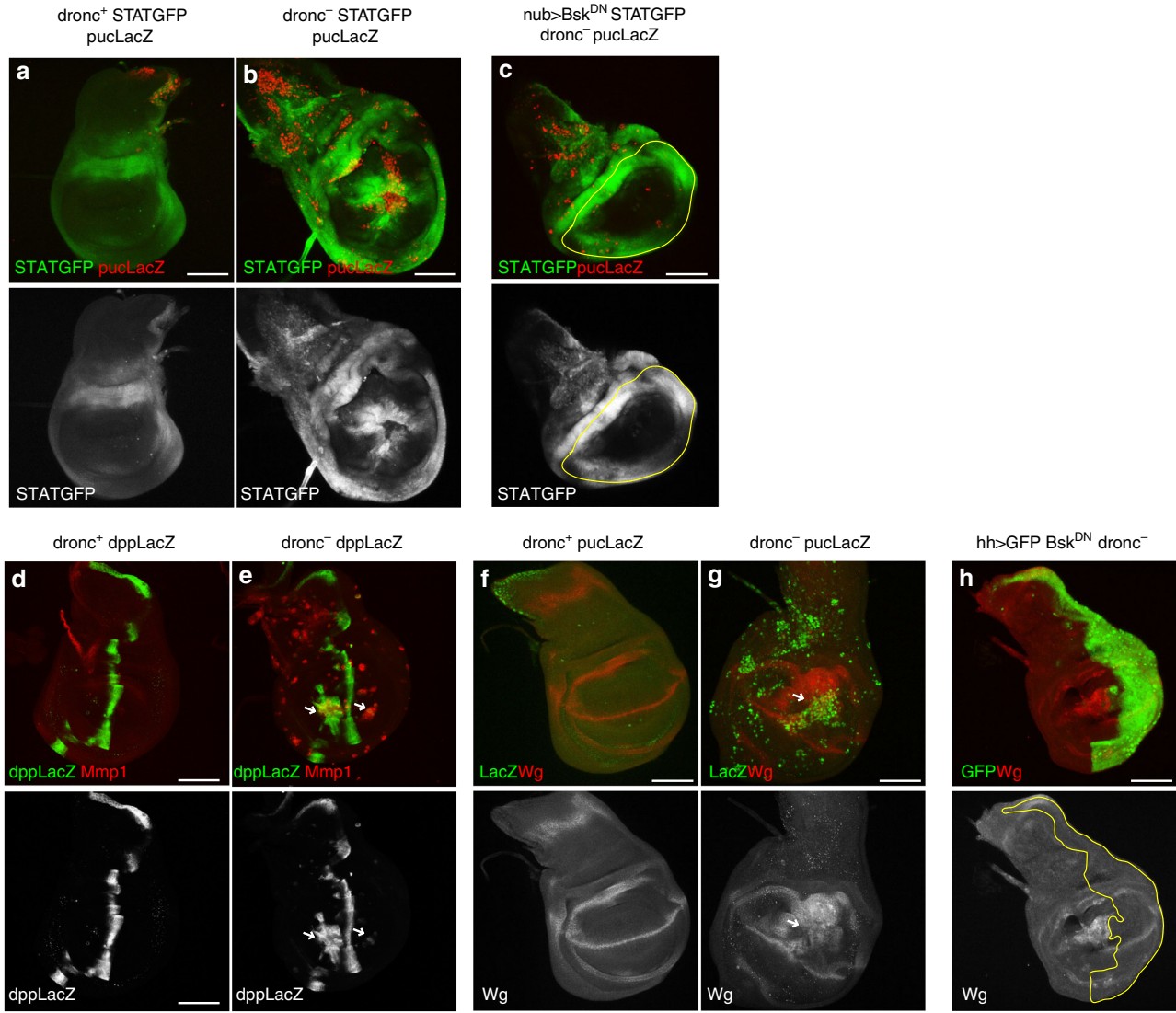

**Fig. 3** JNK persistent activity induces sustained activity of the JAK/STAT, Dpp and Wg pathways. **a**–**c** Images of wing discs of genotypes: **a** *STAT-GFP; pucLacZ dronc^{i24}/+* (*n* = 15), **b** *STAT-GFP pucLacZ dronc^{i24}/dronc^{i29}* (*n* = 29), **c** *UAS-Bsk^{DN};nub-Gal4/STAT-GFP;puc-lacZ dronc^{i24}/dronc^{i29}* (*n* = 3), 96 h after irradiation stained for *STAT-GFP* (green) and *β-gal* (red). Note in **b** the ectopic *STAT* expression, especially in the wing pouch, associated with ectopic JNK activity. In **c**, the suppression of JNK, achieved by forcing *bsk^{DN}* in the wing pouch (the domain of *nubbin* outlined in yellow), prevents ectopic *STAT* expression. **d**, **e** Images of wing discs of genotypes: **d** *dpp-LacZ;dronc^{i24}/+*(*n* = 14) and **e** *dpp-LacZ;dronc^{i24}/dronc^{i29}* (*n* = 17) 96 h after irradiation; *dpp* expression is monitored by a LacZ insert at the *dpp* locus (*dpp-LacZ*, green) and JNK function is indicated by Mmp1 staining (red). The *dronc^{+}* disc (**d**) shows wildtype *dpp* pattern, whereas in *dronc^{−}* (**e**) there is ectopic *dpp* expression (arrows) associated with Mmp1 cells. **f**, **g** Images of wing discs of genotypes: **f** *pucLacZ dronc^{i24}/+* (*n* = 9), **g** *puclacZ dronc^{i24}/dronc^{i29}* (*n* = 22) 96 h after irradiation, stained with anti-Wg antibody (red) and *puc-LacZ* (green) to label JNK activity. The *wg* expression pattern is normal in *dronc^{+}* (**f**), but in *dronc−*, there is clear ectopic expression (**g**), especially in the wing pouch (arrow). Note the overall coincidence between the zones of ectopic *wg* expression and of JNK activity. And **h** *UAS-bsk^{DN};UAS-GFP;puc-lacZ dronc^{i24}/ hh-Gal4dronc^{i29}* (*n* = 8) 96 h after irradiation, stained with anti-Wg antibody (red), the suppression of JNK activity by *bsk^{DN}* in the posterior compartment (green) prevents ectopic *wg* expression. Scale bars are 100 μm

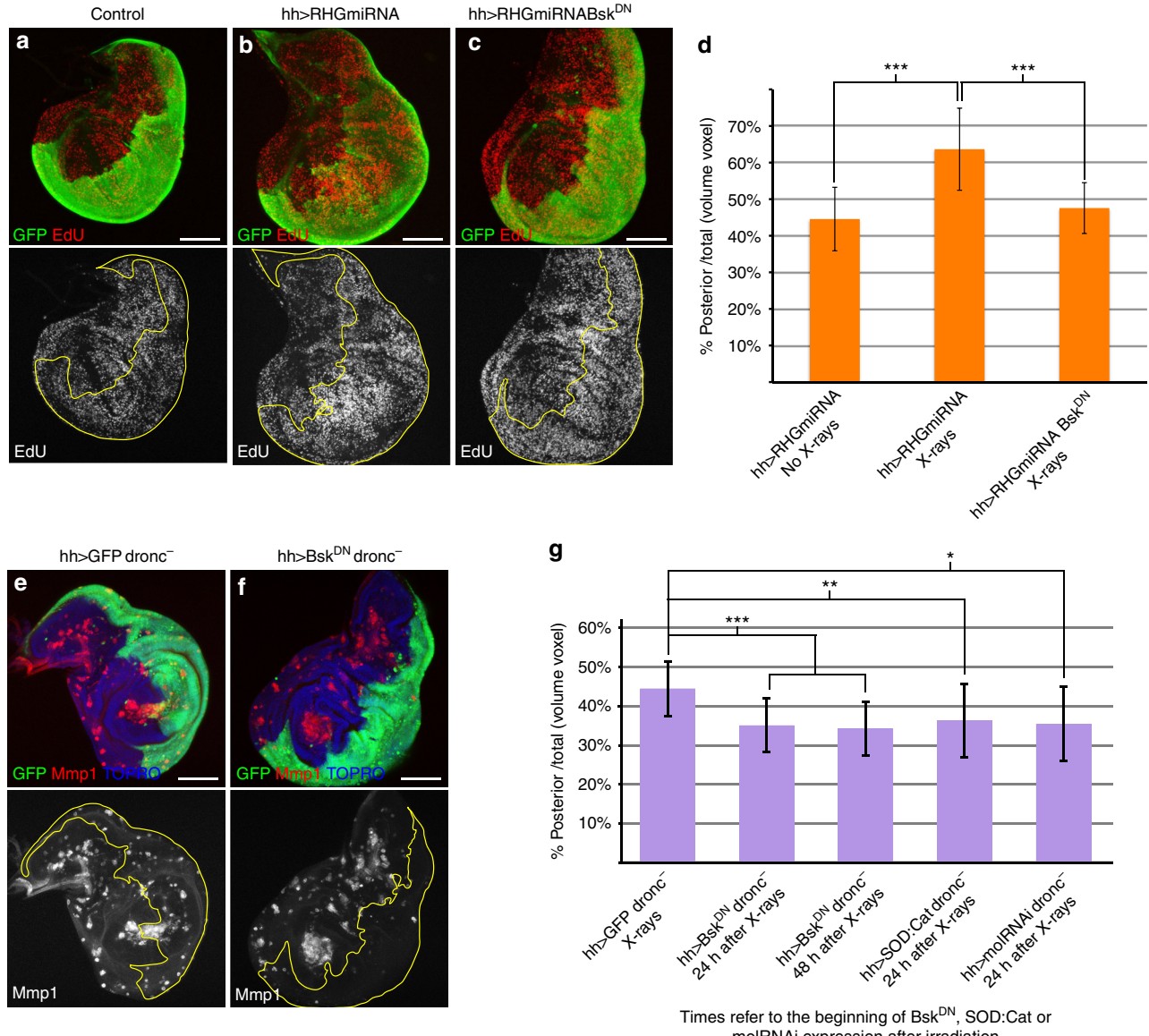

**Fig. 4** Persistent JNK activity causes overgrowths. **a–c** Discs of genotypes: **a** *UAS-GFP;hh-Gal4* ($n = 10$), **b** *UAS-RHGmiRNA/UAS-GFP; hh-Gal4/+*($n = 20$) and **c** *UAS-Bsk^DN; UAS-RHGmiRNA/UAS-GFP; hh-Gal4/+*($n = 11$) 72 h after X-rays. EdU incorporation (red) is increased in the posterior compartment (green) of **b** where apoptosis is suppressed by *UAS-RHGmiRNA*, but when JNK activity is blocked with *Bsk^DN* (**c**), EdU incorporation is similar in both compartments, which is the normal situation (**a**). **d** Quantification of the overgrowth in *UAS-RHGmiRNA/UAS-GFP;hh-Gal4/+*disc ($n = 32$) after irradiation compared to non-irradiated discs ($n = 35$) (***p*-value < 0.0001). Suppression of JNK by *Bsk^DN* ($n = 14$) reverses the posterior compartment to normal size (***p*-value < 0.0001), indicating that JNK is responsible for the overgrowth. **e, f** Discs of genotypes: **e** *UAS-GFPtubGal80^ts/+;dronc^{i24}/hh-Gal4dronc^{i29}* ($n = 10$) and **f** *UAS-Bsk^DN;UAS-GFPtubGal80^ts/+;dronc^{i24}/hh-Gal4dronc^{i29}* ($n = 23$) 96 h after X-rays (see details of the experiment in the Methods section). Discs were stained with Mmp1 (red) to monitor JNK activity and GFP (green) to identify the P-compartment; TOPRO staining delimits the discs. The disc in **e** shows Mmp1 label in both compartments, but expression of the *Bsk^DN* blocks JNK activity (Mmp1, red) as shown in **f**. **g** Quantification of the posterior compartment volume over the total disc volume for the genotypes: *UAS-GFPtubGal80^ts/+; dronc^{i24}/hh-Gal4 dronc^{i29}* (column: hh>GFPdronc⁻ X-rays, $n = 79$), *UAS-Bsk^DN; UAS-GFPtubGal80^ts/+; dronc^{i24}/hh-Gal4dronc^{i29}* expressing *Bsk^DN* 24 h (column: hh>Bsk^DNdronc⁻ 24 h after X-rays, $n = 43$) or 48 h after irradiation (column: hh>Bsk^DNdronc⁻ 48 h after X-rays, $n = 18$). There is a significant size decrease when *Bsk^DN* is expressed 24 h and 48 h after irradiation (***p*-value < 0.0001), indicating that inhibiting JNK after irradiation suppresses overgrowth. Quantification for the genotypes: *UAS-GFPtubGal80^ts/UAS-SOD:UAS-Cat;dronc^{i24}/hh-Gal4dronc^{i29}* (column: hh>SOD:Cat dronc⁻ 24 h after X-rays, $n = 14$) inducing the expression of *SOD:Cat* 24 h after irradiation and *UAS-GFPtubGal80^ts/UAS-molRNAi; dronc^{i24}/hh-Gal4dronc^{i29}* (column: hh>molRNAidronc⁻ 24 h after X-rays, $n = 7$) inducing the expression of *molRNAi* 24 h after irradiation. There is a significant size decrease after the expression of *SOD:Cat* and the inhibition of *mol* (***p*-value < 0.001 and *p*-value < 0.02, respectively), indicating that ROS levels are important to maintain JNK overgrowth. Columns in the graphs represent mean ± s.d. Statistical test used: Mann–Whitney *U*-test. Scale bars are 100 µm

overgrowths depends on JNK activity: in irradiated *hh>RHGmiRNA* discs the overgrowth and increase of EdU incorporation in the posterior compartment is suppressed if they contain the dominant negative form of Bsk (Fig. 4c, d).

To test if the overgrowth observed in the *RHGmiRNA* experiment is induced by the sustained expression of JNK, we made use of the *tub-Gal80^ts* transgene to suppress the JNK pathway with *Bsk^DN* 24 and 48 h after the irradiation in the

posterior compartment of *dronc⁻* mutants. In the 24 h experiment, we already found a significant reduction of the overgrowth in the posterior compartment (Fig. 4f, g), which is also observed in the 48 h experiment (Fig. 4g). Along the same lines, reduction of ROS levels or of *mol* function 24 h after the irradiation also results in suppression of the overgrowths (Fig. 4g). Overall, these results indicate that the overgrowths observed are dependent on persistent JNK activity.

**Lack of apoptosis in *ras^{V12}* tissue leads to tumorigenesis.** Overexpression of the Ras-MAPK pathway is associated with tumour development in vertebrates and *Drosophila*[34–38]. Since it renders cells refractory to stress-induced apoptosis[24–26], we

surmised that this property might confer a tumorigenic potential to cells with elevated levels of Ras activity.

In *Drosophila*, the overexpression of Ras does not generate significant overgrowth, although it up regulates the oncogene *dMyc*[39,40]. In our experiments, the growth of clones or body regions with constitutive Ras function (induced by the *ras^{V12}* transgene) is like in the wildtype: the size of the Sal domain (a region covering 15% of the wing disc) is similar in *sal^{Epv}>GFP* (control) and *sal^{Epv}>ras^{V12}* discs (Supplementary Fig. 2 a–d), and the growth rate of clones containing *ras^{V12}* is also similar to that of the surrounding tissue (Supplementary Fig. 2e). We confirmed that *ras^{V12}*-expressing cells show high levels of *dMyc*[41] (Supplementary Fig. 2d, e). We also confirmed that those cells

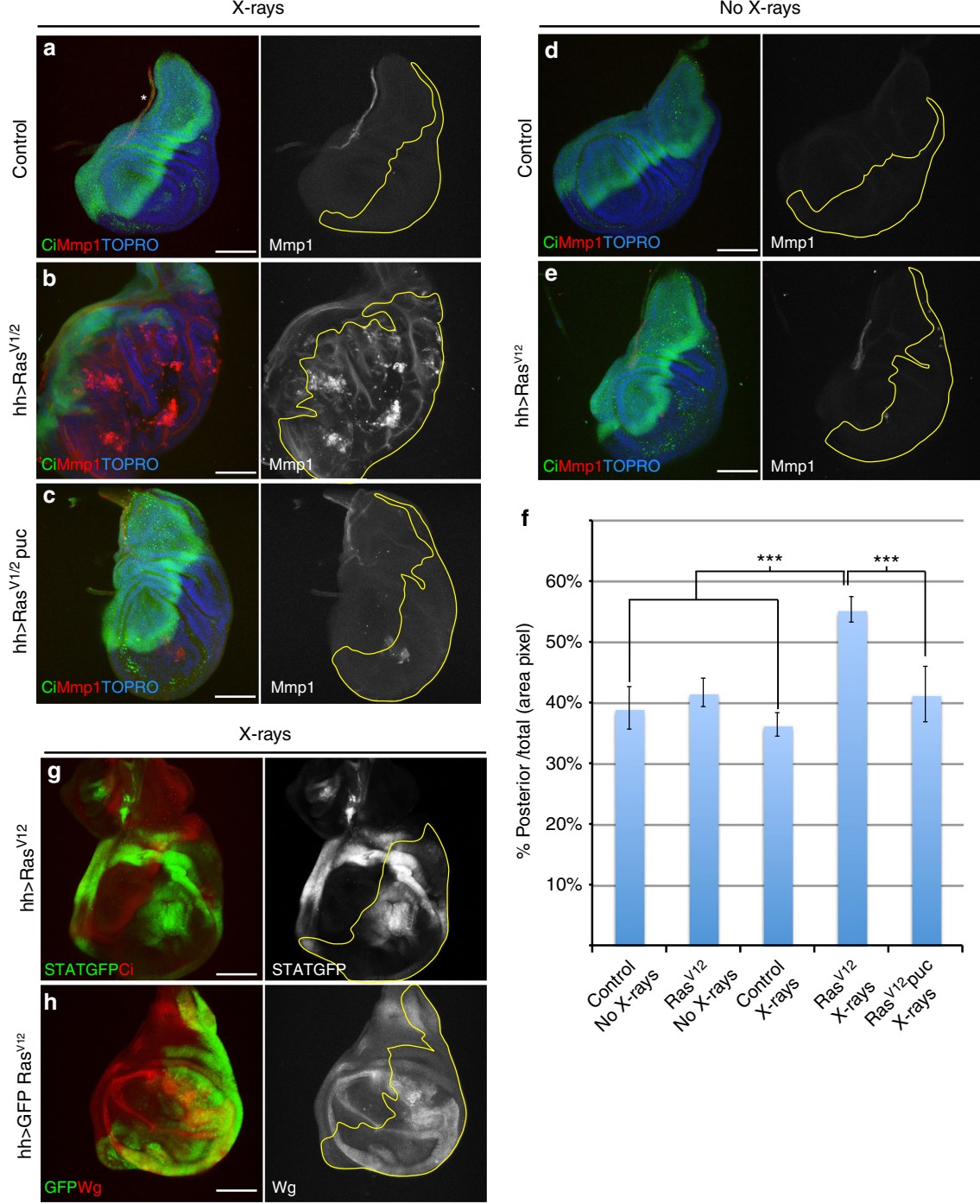

exhibit a reduced apoptotic response to X-rays: the apoptotic levels after irradiation of the posterior compartment or the Sal domain containing $Ras^{V12}$ are much lower than in controls (Supplementary Fig. 3a–c).

We then checked whether the lack of apoptotic response of $ras^{V12}$-expressing cells also causes sustained JNK activity after irradiation. Indeed, in $sal^{Epv}>ras^{V12}$ (Supplementary Fig. 3d) and in $hh>ras^{V12}$ discs (Fig. 5b), JNK remained active in the Sal domain and the posterior compartment, respectively, 72 and 96 h after irradiation. Irradiated (Fig. 5a) and non-irradiated (Fig. 5d, e) controls show no JNK activity. Moreover, Ras activity is associated with ectopic activity of the JAK/STAT and Wg pathways (Fig. 5g, h) and with overgrowth of the posterior compartment (Fig. 5f). These overgrowths are likely the result of the inactivation of the Hippo pathway[42] and require JNK activity, for in irradiated $hh>ras^{V12}$ UAS-puc discs the overgrowth is much reduced (Fig. 5c, f). This observation links the tumorigenic property of the Ras pathway with JNK activity. Interestingly, Ras-induced oncogenic transformations in vertebrates also require c-JNK activity[43].

Since in $ras^{V12}$ cells the apoptotic response is not completely suppressed, we studied the growth of $ras^{V12}$ tissue also mutant for $dronc$. After irradiation, $sal^{Epv}>ras^{V12}$ $dronc^-$ discs showed very large overgrowths in comparison with non-irradiated discs of the same genotype (Supplementary Fig. 3e, f). In this experiment, we introduced a lineage-tracing cassette (see Methods) to follow the progeny of $ras^{V12}$-expressing cells, which expand to the surrounding tissue. It is worth emphasising that these tumorigenic features are triggered by a single 3000 R irradiation administered 96 h before. In these discs, we compared the relative growth of the $ras^{V12}$ tissue with that of the rest of the disc. As shown in Supplementary Fig. 3F, the $ras^{V12}$ cells generate a disproportionate part of the overgrowth. We believe that the strong tumorigenic character of irradiated $ras^{V12}$ cells defective in apoptosis is caused by the sum of two factors: (1) persistent tumorigenic JNK activity and (2) the high metabolic activity conferred by the high $dMyc$ levels[40,44,45].

## Discussion

JNK is associated in vertebrates and in *Drosophila* both with anti-tumorigenic and pro-tumorigenic roles. The anti-tumour function of the single Jun kinase of *Drosophila* is achieved by its ability to induce apoptosis in oncogenic cells[3,4]. In our experiments, we have characterised the pro-tumorigenic function of JNK in apoptosis-suppressed cells by examining the response to a brief induction of JNK caused by X-rays or by a short pulse of $p53$, or of JNK itself. We find that it causes sustained activity of JNK and of genetic factors, e.g. the JAK/STAT, Wg and Dpp pathways, which function downstream of JNK. Furthermore, it gives rise to the appearance of tumorous overgrowths caused by over-proliferation of the apoptosis-defective cells. We emphasise the differences between these experiments and those reported after blocking cell death in apoptotic cells with the baculovirus protein P35[9,33]. Those "undead" cells persistently express JNK, Dpp and Wg, which generate tissue overgrowths. However, in undead cells, the apoptosis program remains fully active, except for the blocking of the effector caspases, and the continuous expression of Dronc generates persistent JNK function[28,46]. In contrast, in the present experiments, the apoptosis program is completely suppressed.

The fact that a short-term activation event is sufficient to induce persistent JNK activity in apoptosis-defective tissues defines an important property of the pathway: it includes a self-sustenance mechanism. In normal tissues open to apoptosis, this feature is largely irrelevant because the cells die soon after JNK activation, but in cells refractory to apoptosis, the function of JNK and of subsidiary pathways is maintained for the rest of the development of the disc. The result is the production of tumour outgrowths. A key aspect of this process is the mechanism of JNK sustenance after the initial event. Based on the results of Khan et al.[30] and our own, we propose a simple model: the activation of JNK (by X-rays, or P53 or Hep itself) induces $mol$ function, which generated high ROS levels that in turn induce JNK function. As $mol$ is induced downstream of JNK[30], it generates an amplification loop that maintains indefinite activity of JNK.

Our results also bear on the tumorigenic properties of the Ras pathway, whose deregulated expression is associated with cancer development in mammals[21]. We show that the principal tumorigenic feature of overexpressing Ras in *Drosophila* is that it makes cells refractory to apoptosis; these cells can acquire irrepressible JNK activity after an irradiation event that would have eliminated them but for the overexpression of Ras. It is of interest that Ras activity is known to have anti-apoptotic activity in mammalian cells[26], although the apoptotic response to irradiation has not been studied. It is also worth noting that oncogenic transformations associated with de-repressed function of Ras in vertebrates require activity of the c-Jun pathway[43]; thus both in *Drosophila* and in vertebrates, tumour development appears to require combined activities of the Ras and JNK pathways.

In their review of Cancer, Hanahan and Weinberg[21] state that one principal feature of cancer cells is that they can evade apoptosis. We would argue the reverse: it is the cells that evade apoptosis that become oncogenic; a simple stress to apoptosis-deficient tissue may have tumorigenic consequences.

**Fig. 5** Constitutive expression of the Ras-MAPK pathway causes overgrowths due to permanent activity of the JNK pathway after irradiation. Images of wing discs of genotypes: **a** control $tubGal80^{ts}/+;hh$-Gal4/+ (n = 13), **b** UAS-$Ras^{V12}/tubGal80^{ts};hh$-Gal4/+ (n = 9) and **c** $tubGal80^{ts}/UAS$-$Ras^{V12};hhGal4/UAS$-$puc^{2A}$ (n = 11) 72 h after irradiation. Non-irradiated wing discs of genotypes: **d** control $tubGal80^{ts}/+;hh$-Gal4/+ (n = 9) and **e** UAS-$Ras^{V12}/tubGal80^{ts};hh$-Gal4/+ (n = 10). Anterior compartments are stained with Ci (green). JNK activity is monitored by Mmp1 (red). Staining with TOPRO facilitates delimiting the discs. Note in **b** the large overgrowth of the posterior compartment that contains UAS-$Ras^{V12}$, associated with ectopic permanent JNK activity. Much of the overgrowth is prevented (**c**) when JNK function is compromised by overexpressing $puc$. There is no Mmp1activity (red) in non-irradiated discs (**d, e**). **f** Quantification of the posterior compartment area (in percentage) over the total disc area in the genotypes: non-irradiated $tubGal80^{ts}/+;hh$-Gal4/+control (column: control No X-rays, n = 12), $tubGal80^{ts}/UAS$-$Ras^{V12};hh$-Gal4/+(column: $Ras^{V12}$ No X-rays, n = 6), and irradiated $tubGal80^{ts}/+;hh$-Gal4/+control (column: control X-rays, n = 8), $tubGal80^{ts}/UAS$-$Ras^{V12};hh$-Gal4/+(column: $Ras^{V12}$ X-rays, n = 9) and $tubGal80^{ts}/UAS$-$Ras^{V12};hhGal4/UAS$-$puc^{2A}$ (column: $Ras^{V12}$ puc X-rays, n = 34). There is a statistically significant size increase of the posterior compartment after irradiation when $Ras^{V12}$ is expressed, compared with non-irradiated discs and with irradiated controls (***p-value < 0.0004, Mann–Whitney U-test). However, if JNK activity is suppressed by overexpressing $puc$, the size of the posterior compartment significantly decreases (***p-value < 0.0001, Mann–Whitney U-test). Columns in the graph represent mean ± s.d. Images of wing discs of genotypes: **g** STAT-GFP/$tubGal80^{ts};UAS$-$Ras^{V12}/hh$-Gal4 (n = 20) and **h** UAS-GFP,$tubGal80^{ts}/UAS$-$Ras^{V12};hh$-Gal4/+ (n = 9) 72 h after irradiation; **g** shows the ectopic expression of STATGFP (green) only in the posterior compartment (delimited by the lack of Ci in red) where $Ras^{V12}$ is expressed; **h** shows the ectopic expression of Wg (red) only in the posterior compartment (green) where $Ras^{V12}$ is active. Scale bars are 100 μm

## Methods

**Drosophila strains.** The *Drosophila* stocks used in this study were: *dronc$^{i29}$*, *dronc$^{i24}$* (A Bergmann, MD Anderson Center, Houston, TX, USA); *puc-lacZ* line (*puc$^{E69}$*)[47], *hh-Gal4* and *en-Gal4* (gift of T Tabata, IMBC, Tokyo, Japan), *nub-Gal4* (our own lab), *sal$^{Epv}$-Gal4* (gift from J. F. de Celis, CBMSO, Madrid, Spain), *tub-Gal80$^{ts}$*[48], *UAS-RHGmiRNA*[49], *STAT-10xGFP*[50], *act>stop>lacZ* (gift of G. Struhl, Harvard Medical School, Boston, MA, USA), *UAS-SOD:UAS-Cat*[29], TRE: RFP[51], *UAS-GFPDbox*[27], *M(3) 67C FRT2A Ubi-GFP, UAS-Flp, UAS-GFP, Df (3L)H99, UAS-hep$^{CA}$, UAS-p53.EX, P(dpp-lacZ), UAS-Bsk$^{DN}$, UAS-puc$^{2A}$, UAS-molRNAi, mol-LacZ* and *UAS-Ras85DV*[12] (Bloomington Drosophila Stock Center).

**Imaginal discs staining.** Immunostaining was performed as described previously[28]. Images were captured with a Leica (Solms, Germany) DB5500 B confocal microscope. The following primary antibodies were used: rabbit anti-Dcp1 (Cell Signalling, antibody #9578) 1:200; mouse anti-β -galactosidase (DSHB 40-1a) 1:50; rabbit anti-β-galactosidase (ICN Biomedicals) 1:2000; mouse anti-Wingless (DSHB 4D4) 1:50; mouse anti-Mmp-1 (DSHB, a combination, 1:1:1, of 3B8D12, 3A6B4 and 5H7B11) 1:50; rat anti-Ci antibody (DSHB 2A1) 1:50; rabbit anti-PH3 (Millipore) 1:100 and guinea pig anti-Myc (our own lab) 1:100. Fluorescently labelled secondary antibodies (Molecular Probes Alexa) were used in a 1:200 dilution. TO-PRO3 (Invitrogen) was used in a 1:600 dilution to label the nuclei.

DHE staining was performed as in Owusu-Ansah et al.[31] with modifications. Larvae were dissected in 1× PBS and incubated with DHE (ThermoFisher Scientific, catalogue number D1168) 30 µM for 5 min at room temperature followed by three washes of 5 min in 1× PBS. Discs were mounted in 1× PBS and images were captured with Leica (Solms, Germany) DB5500 B confocal microscope.

All the *n* numbers stated in the text represent individual discs of the mentioned genotypes. All experiments are replicated at least three times.

**p53/JNK short pulse experiments.** Larvae of the genotypes: *UAS-hep$^{CA}$ /UAS-GFP tubGal80$^{ts}$;hh-Gal4 dronc$^{i29}$/+, UAS-hep$^{CA}$ /UAS-GFP tub-Gal80$^{ts}$; hh-Gal4 dronc$^{i29}$/dronc$^{i24}$, UAS-p53 /UAS-GFP tub-Gal80$^{ts}$; hh-Gal4 dronc$^{i29}$/+* and *UAS-p53 /UAS-GFP tub-Gal80$^{ts}$; hh-Gal4 dronc$^{i29}$/dronc$^{i29}$* were raised at 17 °C for 7 days, then changed to 29 °C for 16 h to activate the corresponding transgene and subsequently changed back to 17 °C. After 96 h, they were dissected and the wing imaginal discs stained with the appropriate antibodies.

**Experiments to suppress JNK.** To block JNK activity, the *UAS-Bsk$^{DN}$* transgene was expressed in larvae of the following genotypes: *UAS-Bsk$^{DN}$; tub-Gal80$^{ts}$, UAS-GFP; dronc$^{i29}$ hh-Gal4/dronc$^{i24}$* and *UAS-Bsk$^{DN}$; nub-Gal4/STATGFP; dronc$^{i29}$/dronc$^{i24}$pucLacZ.* Larvae were raised at 25 °C and changed to 29 °C 24 h before 4000 X-rays dose to enable the expression of *Bsk$^{DN}$*. Irradiation was performed with an X-ray machine Phillips MG102. Larvae were kept at 29 °C for 72 h and then dissected and stained with the appropriate antibodies.

We have also used the *UAS-puc$^{2A}$* transgene in the following genotype: *tub-Gal80$^{ts}$/UAS-Ras$^{V12}$; hh-Gal4/UAS-puc$^{2A}$.* The high levels of the negative regulator *puc* suppress most of JNK function. Larvae were raised at 25 °C and changed to 29 °C 24 h before 3000 X-rays dose to enable the expression of *Ras$^{V12}$* and *puc$^{2A}$*. Larvae were kept at 29 °C for 72 h and then dissected and stained with the appropriate antibodies.

**Quantification of the overgrowths.** No statistical methods were used to estimate the sample size, as we use *Drosophila melanogaster*, and because it has a high fecundity, we can use a large number of animals for our experiments. No randomisation or blind experiments were performed.

For quantification in the *RHGmiRNA* experiments, we used larvae of the genotypes: *UAS-RHG-miRNA/+; hh-Gal4 UAS-GFP/+* and *UAS-Bsk$^{DN}$; UAS-RHG-miRNA/+; hh-Gal4 UAS-GFP/+.* Larvae were raised at 25 °C and changed to 29 °C 24 h before X-rays (4000 R) to induce expression of the *Bsk$^{DN}$* and *RHG-miRNA* transgenes. Control non-irradiated *UAS-RHG-miRNA/+; hh-Gal4 UAS-GFP/+*larvae were raised in the same conditions. All larvae were kept at 29 °C for 72 h and then dissected and stained with GFP and TO-PRO3. Stacks were captured with a Leica (Solms, Germany) DB5500 B confocal microscope, and analysed with Fiji[52] voxel counter plugin to obtain the volume of the samples.

To test if the sustained activity of JNK is responsible for the overgrowths, we blocked JNK activity after irradiation as follows. Larvae of the following genotypes: *tub-Gal80$^{ts}$, UAS-GFP; dronc$^{i29}$ hh-Gal4/dronc$^{i24}$* and *UAS-Bsk$^{DN}$; tub-Gal80$^{ts}$, UAS-GFP; dronc$^{i29}$ hh-Gal4/dronc$^{i24}$* were raised at 17 °C and transferred to 29 °C 24 h before irradiation (4000 R) or transferred to 29 °C 24 h or 48 h after irradiation. All larvae were kept at 29 °C for 72 h and then dissected and stained with GFP and TO-PRO3. Stacks were captured with a Leica confocal microscope, and analysed with Fiji[52] voxel counter plugin to obtain the volume of the samples. Some samples were also stained with Mmp1 to check the JNK activity.

To test if ROS and *mol* are involved in promoting overgrowth, we also quantified the volumes of discs overexpressing *SOD:Cat* and *mol*RNAi, as in the previous experiment for the *Bsk$^{DN}$*.

In the *Ras$^{V12}$* experiments, we used larvae of *tub-Gal80$^{ts}$/+; hh-Gal4/+, tub-Gal80$^{ts}$/UAS-Ras$^{V12}$ hh-Gal4/+* and *tub-Gal80$^{ts}$/UAS-Ras$^{V12}$;*

*hh-Gal4/UAS-puc$^{2A}$.* Larvae were raised at 17 °C and changed to 29 °C 24 h before irradiation (3000 R) to allow transgene expression. Non-irradiated *tub-Gal80$^{ts}$/+; hh-Gal4/+*and *tub-Gal80$^{ts}$/UAS-Ras$^{V12}$ hh-Gal4/+*larvae were raised in the same conditions. All larvae were kept at 29 °C for 72 h and then dissected and stained with rat anti-Ci 1:50 and TO-PRO3. Stacks were captured with a Leica confocal microscope, and analysed with Fiji to obtain the area of the samples.

Statistical analysis was performed with the software Prism5 (Graphpad).

**EdU incorporation.** Wing discs were cultured in 1 mL of EdU labelling solution for 20 min at room temperature and subsequently fixed in 4% paraformaldehyde for 30 min at room temperature. Rabbit anti-GFP (Invitrogen) 1:200 antibody was used overnight at 4 °C before EdU detection to protect GFP fluorescence. EdU detection was performed according to the manufacturer instructions (Click-iT EdU Alexa Fluor 555 Imaging Kit, ThermoFisher Scientific).

**Lineage tracing experiments.** To follow the lineage of cells after irradiation, we performed the following experiment. Larvae of the genotypes: *UAS-GFPDbox/ act>stop>lacZ; puc-Gal4/UAS-flp dronc$^{i24}$* and *UAS-GFPDbox/act>stop>lacZ; dronc$^{i29}$ puc-Gal4/UAS-flp dronc$^{i24}$* were irradiated (4000 R) and kept at 25 °C for 24 h or 72 h before dissection and staining.

In the experiments in which we overexpressed the Ras pathway in the Sal domain of *dronc* mutant larvae, e.g. *sal$^{Epv}$-Gal4, UAS-GFP/act>stop>lacZ UAS-Flp; dronc$^{i29}$ UAS-Ras$^{V12}$/dronci24*, the *flippase* activity directed by the *sal$^{Epv}$-Gal4* driver induces recombination of the *act>stop>lacZ* cassette in all the cells of the Sal domain. These cells become irreversibly labelled with *lacZ*, thus allowing the identification of the cellular progeny of the domain.

**Data availability.** The authors declare that the data supporting the findings reported in this manuscript are available within the article and the Supplementary Information files or from the corresponding author upon request.

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

## Acknowledgements
We thank the members of our lab for comments and discussions and Angélica Cantarero and Rosa Gonzalez for technical help. We also thank Jose F. de Celis for comments and suggestions. This work has been supported by grants BFU-2015-67839-P of the Ministerio de Economia y Competitividad and the XVII Concurso Nacional of the Fundación Ramón Areces.

## Author contributions
N.P., M.M. and G.M. conceived the project, designed the study and analysed the data. N.P., M.M. and I.M. performed the experiments. N.P. and G.M. wrote the manuscript. All authors edited and approved the final manuscript.

## Additional information

**Competing interests:** The authors declare no competing interests.

