## [Peer Review File · Nature Communications]

Reviewers' comments:

Reviewer #1 (Remarks to the Author):

The manuscript "Short-term activation of the Jun N-terminal Kinase pathway in apoptosis-deficient cells of *Drosophila* induces tumorigenesis" by Pinal et al. is an interesting extension of our understanding of the interaction between JNK signaling and other signals under circumstances of tissue stress/damage (e.g. irradiation) in the wing imaginal disc. The fact that the combination of active JNK and Ras signaling induces overgrowth is not new, nor is the fact that JNK signaling activates JAK/STAT and Wg signaling. The authors claim that the novelty of the paper lies in the finding that a short burst of JNK signaling leads to sustained JNK signaling in cells that fail to die. However, this aspect of the story is incomplete and not yet convincing, and several questions remain to be addressed before it would be ready for publication.

It is appreciated that the authors use multiple methods to block apoptosis that avoid generating "undead" cells, which would confound the analysis. Similar careful experimental design should be able to address the questions listed below.

The main conclusion of the paper is that when cells with activated JNK signaling are prevented from dying, they sustain JNK signaling through some positive feedback mechanism. However, it is not clear that this conclusion can be drawn for all JNK-expressing cells in the wing disc. There are many examples of cells temporarily activating JNK, either during development or in response to tissue damage, and not dying. For example, JNK signaling is activated in the leading edge cells during dorsal closure and in border cells during oogenesis. In the wing disc, JNK signaling is activated during wound closure and during metamorphosis. These cells that experience JNK signaling neither undergo apoptosis nor express JNK permanently. Thus, more information is required and the model must be revised to account for such instances.

1. After irradiation, scattered cells temporarily express puckered (Fig. 1). The puckered-expressing cells later disappear, and the authors assume they have all undergone apoptosis, but it is not clear whether they have all died or whether some have remained in the epithelium and stopped expressing puc. Lineage tracing would answer this question, either using a system like G-trace with puc-GAL4, or a flip-out system as used in Bosch et al., 2008.

2. How is JNK sustaining its own activation? Are ROS levels elevated in these cells? Identification of the mechanism would increase the impact of the work significantly.

3. 1E and 1G look quite different from each other, so JNK signaling is not active in the same cells in the same pattern at 96 hr as it was at 24 hr. At 96 hr there appear to be clumps of cells, not uniformly scattered cells as at 24hr, that express puc-lacZ. Are these clumps of cells still in the epithelium or are they being extruded? What happened to the cells that were puc-lacZ at 24 hr that did not go on to generate a clump of puc-lacZ cells at 96 hr? Since apoptosis was blocked, were they extruded without undergoing apoptosis? Did they stop expressing puc-lacZ but remain in the epithelium? Did they migrate to cluster together, generating the appearance of fewer groupings of puc-lacZ cells at 96 hr?

4. It has been shown that hinge cells are resistant to apoptosis after irradiation. Why then does puc-lacZ not persist in the hinge region of the irradiated *dronc*⁺ discs? According to the model proposed here, the hinge cells should have persistent JNK activation, but they do not.

5. Activating JNK signaling with hepCA results in very strong activation that may not mimic physiological levels of activation (Fig. 2). What happens to JNK signaling when it is activated to endogenous levels in *dronc*⁻ animals – for example in dorsal closure or after a cut to the wing disc?

6. Data presented in Fig S1 should be quantified: the size of the Sal domain, mitotic index from PH3 staining, Myc immunofluorescence levels, etc.

7. The authors propose that rasV12 cells that don't die are tumorigenic in part due to "high metabolic activity conferred by high dMyc levels". This hypothesis could be tested by reducing Myc levels in the disc and quantifying the effect on overgrowth and/or metabolism. (The paper cited to support this statement, Prober and Edgar Genes Dev 2002, does not appear to demonstrate changes in metabolism in response to changes in Myc levels.)

Minor points

1. Is RX an accepted abbreviation for irradiation?
2. In Figure S1 the Myc staining has very high background and is difficult to see, especially in panel E.
3. Imaginal discs in the figures should all be oriented in the same way – dorsal up anterior left is common, but whichever orientation is chosen it should be the same in each figure panel.

Reviewer #2 (Remarks to the Author):

This manuscript by Gines Morata's group investigates the fate of apoptosis-deficient cells after short-term activation of JNK in *Drosophila*. They show that a pulse of JNK activation triggered by X-radiation, p53 expression, or JNKK activation persists for the rest of the development in the wing disc cells deficient for apoptosis (*dronc* mutants, H99 mutants, or RHGmiRNA-expressing cells). Such persistent JNK-activating cells show upregulation of JNK-targets *wg*, *dpp*, and *STAT* signaling, resulting in tissue overgrowth. Similar results were also obtained when apoptosis was blocked by oncogenic RasV12 expression, arguing that cells evading apoptosis become tumorigenic due to the acquisition of permanent JNK activity.

The authors' finding that a pulse of JNK activation leads to persistent JNK activation for at least 96 hrs after the pulse when apoptosis is blocked is very interesting and novel. It is also an interesting concept that cells resistant to apoptosis become tumorigenic because they acquire permanent JNK activity after stress events. However, in this manuscript, the mechanism by which JNK is persistently activated for such a long period is totally unknown.

Specific comments:

1. The novelty in this manuscript relies on the phenomenon that transient activation of JNK can persist for a long period if apoptosis is inhibited, given that JNK-dependent tissue overgrowth or Ras-mediated apoptosis inhibition has already been reported. The phenomenon itself is very interesting, but the authors do not provide any mechanisms underlying the persistent JNK activation (without any triggers). Thus, adding more data that suggest the mechanism of how persistent JNK activation is achieved would significantly improve the manuscript.
2. The authors show that JNK activation triggered by irradiation disappears by 96 hrs using the *puc-lacZ* reporter, which makes the whole animal heterozygous mutant for *puc*. Because *puc* is JNK phosphatase, *puc/+* background would sensitize JNK activity, which could be the reason why strong JNK activation persists for a long period. To clarify this, the authors should perform the same experiments shown in Fig. 1B-G using other JNK activity reporters such as MMP1 or TRE-

red.

3. In Fig. 2, the background of *dronc*⁺ controls (A, C) is too low (almost no background) compared to *dronc*⁻ samples (B, D; much higher). The background of all the experiments should be adjusted to similar level.

4. The authors argue that oncogenic Ras causes JNK-activated cells to become tumorigenic because Ras activity blocks apoptosis. However, it has been reported that activation of Ras and JNK signaling in *Drosophila* imaginal disc causes tumorigenesis via inactivation of the Hippo pathway (Ohsawa et al., Nature, 2012), which should also be mentioned.

5. The authors claim that they used *dronc* mutants to completely suppress the apoptotic program so that they can make difference between the current work and the previous work that used the effector caspase inhibitor p35, which generates undead cells. However, the authors also used the RHGmiRNA and H99 mutation, which would cause the same effect as p35. This is confusing and needs to be clarified.

6. The authors say "permanent" JNK activation, yet they only examine 96 hrs after the pulse at longest. It might be better to say "persistent" or "sustained" JNK activation.

Reviewer #3 (Remarks to the Author):

Drosophila has provided the premier model system for examining how cells and tissues respond to insults including irradiation or physical damage, to allow regeneration. Much is known about the molecular pathways involved, which include activation of the JNK pathway and subsequent induction of downstream events including apoptosis and activation of the Wnt and Dpp signaling pathways. Here the authors explore an interesting follow-up question: while JNK activation normally triggers apoptosis, what happens with respect to JNK activity if apoptosis is blocked. Once again, we know some things-e.g., undead cells are known to overgrow in response to damage. Here the authors present evidence suggesting that JNK activity may be sustained if apoptosis is blocked, and that this can contribute to overgrowth. They include some interesting data. The authors conclude that permanent/prolonged JNK activation (in apoptosis deficient cells) is an important component of the JNK pathway's role in tissue overgrowth, however, there are some significant limitations in their support of this conclusion. (1) The experiments in the current manuscript that purport to show that JNK is required for tissue overgrowth do not fully demonstrate that it is the sustained activity of JNK that is required. That would require manipulating JNK activity temporally (e.g., turning JNK off after 48hrs), and measuring the effects on tissue overgrowth, which these experiments do not do. They block any JNK response, including the initial response, which is already known to be required for tissue overgrowth following damage to apoptosis deficient cells. (2) The data demonstrating "permanent" JNK activation are confounded by potential stability of the protein markers used as JNK reporters. (3) There is less mechanistic insight into what happens to trigger persistent JNK activation than might be expected. These and other major concerns are detailed below. Given this, the current observations appear to represent a relatively modest advance in our current understanding of JNK activity in apoptosis deficient cells, the subsequent JNK-induced signaling events, and their combined effects on proliferation. However, if the authors can address the major concerns detailed below, which would require quite a lot of work, the impact of the work would be more substantial.

Major points:

(1) Is activity truly sustained? The use of MMP1 or *puc-lacZ* as a marker of continued JNK activity seems problematic since MMP1 or *lacZ* may be stable proteins. They are blocking apoptosis of the cells that turn on JNK at 24 hours. Thus, the presence of MMP1 or *lacZ* protein at day 4 in these

genotypes could simply reflect earlier JNK activity, not ongoing JNK activation. What is known about turnover rates for these proteins? A better marker for these experiments would be a destabilized GFP driven by a JNK-inducible promoter/Gal4 (e.g., *puc-Gal4*[E69]; Pastor-Pareja et al. *Dev Cell*. 2004), or, as the current manuscript notes that JAK/STAT is also a JNK target, perhaps one could use the publically available JAK/STAT reporter that expresses destabilized GFP (10xSTAT-dGFP; Bach et al., *Gene Expr Patterns*. 2006).

(2) A very recently published study (Khan et al., *PLoS Genetics*. July 2017) has demonstrated that, following a damage event in the wing disc, JNK signaling is activated and creates a positive feedback loop through upregulation of the DUOX maturation factor, *moladietz*. This maintains JNK activity throughout regeneration, days after the damage occurred. These observations unfortunately undermine the novelty of some observations in the current manuscript (permanent JNK activation following transient injury), though the current study does demonstrate that simply activating JNK without cellular/tissue damage is sufficient to maintain its activity. While this is an important new observation, the other work should be cited and this work put into that context.

(3) The current story is somewhat minimal. The first three of five figures all make a similar point—transient cellular insult/activation of JNK signaling leads to prolonged activation of JNK and its downstream effector pathways. As all of these observations are fundamentally an extension of known relationships: damage activates JNK, p53 can activate JNK, JNK activates JAK-STAT, *Wg*, and *Dpp*—the novelty is the extended time over which the response is maintained if apoptosis is blocked. To have three main figures dedicated to this seems to be a bit excessive. Some data could likely be combined to a single figure and some moved to Supplementary. Moving some data to Supplemental may reduce already fairly sparse figures, though that might be offset by the addition of graphs showing the quantification of various phenotypes and additional controls, as noted below.

(4) The authors do not speculate on the mechanism by which JNK activity might remain high days after the initial stimulus. It seems worth at least mentioning potential mechanisms, and for this level of journal, testing one or a few. For example, the recent Khan et al., (2017) study demonstrated that, following damage to the wing disc, JNK signaling is activated and creates a positive feedback loop through upregulation of the DUOX maturation factor, *moladietz*. This appears to maintain JNK activity throughout regeneration, days after the damage occurred. It would be interesting to test if *moladietz* and ROS are involved in the permanent JNK activation following the various treatments presented in the current study. In damaged eye discs, macrophages are recruited to the disc and can activate JNK signaling in the eye disc cells (Fogarty et al., *Science Rep*. 2013). Could a similar mechanism be at play here? Are macrophages recruited to wing discs following transient JNK activation? Since JNK is known to regulate expression of many metabolic genes, maybe the cells are now stressed from having their metabolism perturbed, which in turn keeps JNK on. Obviously there are numerous possibilities, not all of which would need to be followed, but some attempt to define a mechanism would significantly increase the impact of this work.

(5) The experimental design does not seem to allow one to distinguish between the role of initial JNK signaling and prolonged JNK signaling, since the expression of *bskDN* or *puc* occurs throughout, and even precedes, the transient injury. It is known that JNK signaling is required for overgrowth following injury in apoptosis deficient cells (Ryoo et al., *Dev Cell*. 2004). Experiments need to be designed that allow for the initial JNK activation, but then turn JNK off after some later time point to clearly test the role of the late JNK activity. The controls used likely do not accomplish this, as in a wildtype background, cells with initial high levels of JNK activation die, and when the experimenters do block both apoptosis and JNK activity using *BskDN*, they do it from the start of the experiment. Without this, the major novel contention of the paper, that prolonged activity of the JNK pathway (and its downstream effectors) is essential for tissue overgrowth, is not supported by the current data.

(6) The current experiments do not definitively show that permanent activation of JNK (in apoptosis deficient cells) is sufficient to induce tissue overgrowth, because they do not quantitate the effect on growth of transiently activating JNK through misexpression of p53 or hepCA (Figure 2). Quantifying overgrowth in these experiments, using the posterior/anterior ratio estimates used in Figs 4+5, would do this.

Minor points:

There is little quantification of phenotypes. There are sample sizes indicated for each experiment, but then just a single image is shown. For better or worse, the current standard is to quantify the marker in question and present those data graphically or at least as mean \pm SD. For this level of journal, this does not seem unreasonable. There are some situations where a simple "yes or no" result might not require quantification (e.g., Fig 1 or 2), however, the data represented in other figures would benefit from it (e.g., StatGFP expression in Fig 3A-C, EdU incorporation in Fig 4AC).

Figure 1 – Needs separate channels, at least for H-J. Also, it should be made clear in the Figure, not just the legend, that H-J are irradiated discs 96h. There should at least be a control (no irradiation) for these experiments, and also perhaps a 24hr timepoint.

Panel H. Minute⁺ dronc⁻ clones are generated in a M^{+/-} background, however, it is never discussed why the Minute mutation is present, or what consequences that may have on the experiment. Given the role of Minute mutants in cell competition, and the role of JNK signaling in cell competition, it seems that this may affect the results of these experiments. Is the Minute required to generate dronc⁻ clones? Since it is possible to generate homozygous null mutant wing discs (as in Panels C,E,G), why not simply use that approach and include wildtype control discs?

Figure 2 – For both the p53 and hepCA expression experiments, there needs to be an earlier time point showing that JNK was activated in the Dronc⁺ control discs and goes away by 96hrs.

Figure 3 – Panel C, the expression pattern of nubbin-Gal4 needs to be delineated for readers not familiar with the wing disc and this particular driver (at least provide an outline on the image).

Why do panels F,G switch to different color palette? Should maintain red and green for consistency.

Supplemental Fig 2 – In panels A and B, shouldn't the non-GFP area be outlined in yellow, to indicate the region of Gal4 expression where ras is misexpressed? Otherwise it appears that there is more cell death in the region of interest.

Discussion. It might be worth speculating about whether JNK can be activated at sub-lethal doses. If not, this may suggest that there is some threshold, beyond which JNK cannot be turned off.

Pg. 4. "This result suggests that once activated, JNK is able to sustain its own expression." Should be "activity" not "expression".

Pg. 5. Why not show the data that JNK is permanently activated following irradiation of p53 mutant discs?

Pg. 6. "In this experiment, we introduced a lineage tracing cassette (see Methods) to follow the behaviour of rasV12 –expressing cells, which invade the surrounding tissue." The use of "invade" here seems overreaching since it can imply migration, however, these cells may instead result from increased proliferation and subsequent loss of sal-driven GFP. Even in the control (Suppl. Fig 2E), there appear to be sizeable regions of lineage⁺ cells that do not (or no longer) express sal-driven GFP.

There are numerous typographical/spelling errors. The manuscript and figures need to be proofread more carefully.

Response to the referees

In what follows we list the relevant comments by the referees (in italics) and our response

Referee 1

The main conclusion of the paper is that when cells with activated JNK signaling are prevented from dying, they sustain JNK signaling through some positive feedback mechanism. However, it is not clear that this conclusion can be drawn for all JNK-expressing cells in the wing disc. There are many examples of cells temporarily activating JNK, either during development or in response to tissue damage, and not dying.

The referee is right in that JNK signalling is not always associated with cell death, but we deal here with JNK activated by X-rays in the wing disc and in those circumstances it induces apoptosis (see Shlevkov and Morata 2012). We surmise that for example the maintenance of JNK in the proximal region of the wing disc is achieved by some mechanism that prevents in those cells the apoptosis-inducing function of JNK

1. After irradiation, scattered cells temporarily express puckered (Fig. 1). The puckered-expressing cells later disappear, and the authors assume they have all undergone apoptosis, but it is not clear whether they have all died or whether some have remained in the epithelium and stopped expressing puc. Lineage tracing would answer this question, either using a system like G-trace with puc-GAL4,

We agree with the reviewer that it was interesting to follow the lineage of puc-expressing cells, both for those open to apoptosis (*dronc*⁺) and in apoptosis-suppressed cells (*dronc*⁻). Thus following his/her suggestion we have traced the lineage of the cells the gain puc (JNK) expression after the irradiation. The procedure and the results are described in the revised manuscript (pages 4 and 5, Suppl. Fig 1, also in the Methods section). In brief, we find, 1) that in *dronc*⁺ cells the activation of JNK by X-rays does not necessarily result in the death of the cell; some survive and contribute to the reconstituted wing disc. However 72 h after the irradiation none of them express puc, indicating the JNK has been turned off and 2) in *dronc*⁻ cells the majority of those expressing puc cells still retain JNK activity 72 h after the irradiation, as revealed by GFP, for which we used an unstable Dbox form to avoid the possibility of perdurance. These experiments provide another proof of the persistence of JNK activity in apoptosis defective cells. They also show that X-Rays JNK activate JNK to sub-lethal levels.

2. How is JNK sustaining its own activation? Are ROS levels elevated in these cells? Identification of the mechanism would increase the impact of the work significantly.

This is a very important issue mentioned by the three referees, which we have considered very seriously, and have addressed by performing several new experiments, which are included in the revised version.

First, we have examined the possible association of Reactive Oxygen Species (ROS), known to activate JNK, with the maintenance of JNK activity. As shown in the new Fig. 2b there is ROS activity in apoptosis-defective cells 96 h after the irradiation.

Second, we checked the possible involvement of moladietz (mol), which has been recently (Khan et al 2017) shown to induce ROS production and subsequent JNK function. Making use of a mol-LacZ insert we find mol activity in apoptosis-deficient cells 96 h after irradiation (new Fig. 2e).

Third, we have assayed the functional requirements of ROS and mol for the sustenance of JNK by either reducing ROS or suppressing mol function. In both cases there is a strong reduction of JNK activity in apoptosis-deficient cells 96 h after X-Rays (new Fig. 2c,f)

These experiments are now included in the revised version (pages 6 and 9) and we believe strengthen considerably the impact of the work. They provide a plausible model for the sustenance of JNK in apoptosis defective cells.

3. 1E and 1G look quite different from each other, so JNK signaling is not active in the same cells in the same pattern at 96 hr as it was at 24 hr. At 96 hr there appear to be clumps of cells, not uniformly scattered cells as at 24hr, that express puc-lacZ. Are these clumps of cells still in the epithelium or are they being extruded?

What happened to the cells that were puc-lacZ at 24 hr that did not go on to generate a clump of puc-lacZ cells at 96 hr? Since apoptosis was blocked, were they extruded without undergoing apoptosis? Did they stop expressing puc-lacZ but remain in the epithelium? Did they migrate to cluster together, generating the appearance of fewer groupings of puc-lacZ cells at 96 hr?.

We agree with the referee that the patterns of JNK activity look different at 24 and 96 h in Fig. 1. We specifically looked for extrusion of puc-expressing cells but did not find evidence for it. However, the lineage experiment mentioned above (included in the revised version of the manuscript) provides some answers to the referee's questions. Our analysis of the lineage of puc-expressing cells in dronc⁺ discs reveals that not all the cells that acquire puc expression maintain it (Suppl. Fig 1): there are clones of cells labelled with LacZ that do not express GFP. This may explain in part the differences of puc expression patterns at 24 and 96 h. Also local differences in JNK activity along the process may give different patterns of the target puc

4. *It has been shown that hinge cells are resistant to apoptosis after irradiation. Why then does puc-lacZ not persist in the hinge region of the irradiated dronc+ discs? According to the model proposed here, the hinge cells should have persistent JNK activation, but they do not.*

We are aware of the work indicating that hinge cells are resistant to apoptosis (Tamori et al Plos Biol 2016) but do not see a differential response of those cells in our experiments

5. *Activating JNK signaling with hepCA results in very strong activation that may not mimic physiological levels of activation (Fig. 2). What happens to JNK signaling when it is activated to endogenous levels in dronc- animals – for example in dorsal closure or after a cut to the wing disc?*

We agree that inducing hep^{CA} with the Gal4/UAS method may activate JNK at non-physiological levels, but in our experiments hep^{CA} is activated only for a short period of 16 hrs. Moreover, it is known that in experiments of this kind the effective elimination of the Gal80^{TS} protein takes about 8 hrs, thus reducing the time of hep^{CA} activity to about 8 hrs. In the controls, JNK disappears after 24 hrs.

A suggestion by the referee of examining JNK in discs after a cut is impractical and in our view not more rigorous. A damaged disc would have to remain in situ after injuring the larva or being cultured in heterologous medium. Situations prone to create artefacts

6. *Data presented in Fig S1 should be quantified: the size of the Sal domain, mitotic index from PH3 staining, Myc immunofluorescence levels, etc.*

We did not quantify the data in Supplementary fig. 1, now 2, because there are previous reports (Karim and Rubin 1998; Prober and Edgar 2002) dealing with those aspects of Ras overexpression,

7. The authors propose that rasV12 cells that don't die are tumorigenic in part due to "high metabolic activity conferred by high dMyc levels". This hypothesis could be tested by reducing Myc levels in the disc and quantifying the effect on overgrowth and/or metabolism. (The paper cited to support this statement, Prober and Edgar Genes Dev 2002, does not appear to demonstrate changes in metabolism in response to changes in Myc levels.).

The referee was right about the citation of Prober and Edgar. The connection between Myc and cell metabolism is specifically mentioned in Drosophila in Johnston et al 1999. In vertebrates, there also is evidence for the connection between Ras overexpression and

reprogramming of cell metabolism (see for example the recent paper by Dejure and Eilers EMBO J 2017)

Minor point 1 Is XR an accepted abbreviation for irradiation?

It was our mistake. We have modified the notation and use X-rays in the figures to refer to irradiation

Minor point 2. In Figure S1 the Myc staining has very high background and is difficult to see, especially in panel E

We have tried to get pictures with less background, but without much success. dMyc is expressed ubiquitously what makes it difficult to obtain pictures that show variations of Myc levels clearly

3. Imaginal discs in the figures should all be oriented in the same way – dorsal up anterior left is common, but whichever orientation is chosen it should be the same in each figure panel

Agreed. We have modified the disposition of the figures in the revised version

Referee 2

1. The novelty in this manuscript relies on the phenomenon that transient activation of JNK can persist for a long period if apoptosis is inhibited, given that JNK-dependent tissue overgrowth or Ras-mediated apoptosis inhibition has already been reported. The phenomenon itself is very interesting, but the authors do not provide any mechanisms underlying the persistent JNK activation (without any triggers). Thus, adding more data that suggest the mechanism of how persistent JNK activation is achieved would significantly improve the manuscript.

First, we would like to emphasize the interest and novelty of the phenomenon itself, recognised by the referee.

But we also agree that identifying the mechanism(s) underlying the persistent JNK function would increase significantly the impact of the manuscript, a point raised by the three referees. The several experiments performed to address this issue and the modifications introduced in the revised version are described above in the response to referee 1

2. The authors show that JNK activation triggered by irradiation disappears by 96 hrs using the puc-lacZ reporter, which makes the whole animal heterozygous mutant

for puc. Because puc is JNK phosphatase, puc/+ background would sensitize JNK activity, which could be the reason why strong JNK activation persists for a long period. To clarify this, the authors should perform the same experiments shown in Fig. 1B-G using other JNK activity reporters such as MMP1 or TRE-red

As shown in the new Fig. 1h,j we have used Mmp1 to reveal JNK activity in the p53 and hep^{CA} experiments. We also use it in other experiments, see Figures 4 and 5 for example. In the experiments illustrated in Fig 2d,e we have used TRE-RFP

3. In Fig. 2, the background of dronc+ controls (A, C) is too low (almost no background) compared to dronc- samples (B, D; much higher). The background of all the experiments should be adjusted to similar level.

We now present better pictures and with uniform background. Those pictures have now been moved to the new Fig.1h,j

4. The authors argue that oncogenic Ras causes JNK-activated cells to become tumorigenic because Ras activity blocks apoptosis. However, it has been reported that activation of Ras and JNK signaling in Drosophila imaginal disc causes tumorigenesis via inactivation of the Hippo pathway (Ohsawa et al., Nature, 2012), which should also be mentioned.

We mention the paper by Ohsawa et al., 2012 in the revised version. However, we would like to point out any increase in cell proliferation is most likely be mediated by down regulation of the Hippo pathway, for Hippo is the principal regulator of proliferation in the imaginal discs. Thus the down regulation of Hippo, observed during tumorigenesis (see for example Menendez et al 2010) is likely not to be a cause but a consequence of the tumorigenesis.

5. The authors claim that they used dronc mutants to completely suppress the apoptotic program so that they can make difference between the current work and the previous work that used the effector caspase inhibitor p35, which generates undead cells. However, the authors also used the RHGmiRNA and H99 mutation, which would cause the same effect as p35. This is confusing and needs to be clarified

The referee is not right in this issue. Both the RHGmiRNA and the H99 deletion suppress the function of the pro-apoptotic genes, which act upstream dronc. P35 acts directly on the effector caspase Drice, which is downstream Dronc. Thus RHGmiRNA and H99 suppress apoptosis, whereas P35 blocks just the effector caspase Drice and the apoptotic loop remains active

6. The authors say “permanent” JNK activation, yet they only examine 96 hrs after the pulse at longest. It might be better to say “persistent” or “sustained” JNK activation.

Following the suggestion by the referee, we have changed the word “permanent” for “persistent” or “sustained” in the revised version. However, we wish to point out that 96 hrs after X-rays is the longest time we can analyse, for the discs stop growth and initiate pupariation. Thus we could argue that once activated JNK remains permanently active during the remaining of development

Referee 3

Here the authors present evidence suggesting that JNK activity may be sustained if apoptosis is blocked, and that this can contribute to overgrowth. They include some interesting data. The authors conclude that permanent/prolonged JNK activation (in apoptosis deficient cells) is an important component of the JNK pathway's role in tissue overgrowth, however, there are some significant limitations in their support of this conclusion.

(1) The experiments in the current manuscript that purport to show that JNK is required for tissue overgrowth do not fully demonstrate that it is the sustained activity of JNK that is required. That would require manipulating JNK activity temporally (e.g., turning JNK off after 48hrs), and measuring the effects on tissue overgrowth, which these experiments do not do. They block any JNK response, including the initial response, which is already known to be required for tissue overgrowth following damage to apoptosis deficient cells.

Certainly we demonstrate that the overgrowths are dependent on JNK activity. But the referee was right in that we did not demonstrate they depend on sustained JNK activity. Following his/her suggestion we have performed experiments manipulating JNK activity at two time points, 24 and 48 h after irradiation. We have also manipulated the levels of ROS and of mol function at the same time points. The results are presented in Fig. 4fg and demonstrate that sustained JNK function is necessary for the generation of overgrowths

(1bis) Is activity truly sustained? The use of MMP1 or puc-lacZ as a marker of continued JNK activity seems problematic since MMP1 or lacZ may be stable proteins. They are blocking apoptosis of the cells that turn on JNK at 24 hours. Thus, the presence of MMP1 or lacZ protein at day 4 in these genotypes could simply reflect earlier JNK activity, not ongoing JNK activation. What is known about turnover rates for these proteins? A better marker for these experiments would be a destabilized GFP driven by a JNK-inducible promoter/Gal4 (e.g., puc-Gal4[E69]; Pastor-Pareja et al. Dev Cell. 2004), or, as the current manuscript notes that JAK/STAT is also a JNK target, perhaps one could use the publically available JAK/STAT reporter that expresses destabilized GFP (10xSTAT-dGFP; Bach et al., Gene Expr Patterns. 2006).

We have looked carefully at this issue. To overcome the possible problem of the stability of the Mmp1 or lacZ markers, the referee suggested using a destabilized GFP protein driven by puc-Gal4. We have used the unstable GFPDbox (Chen et al Genetics 2015), in the lineage tracing experiment. We find that a large number of dronc⁺ cells express puc-Gal4 24 h after the irradiation and also become labelled with lacZ. Some survive JNK activation and leave lacZ progeny that can be visualised after 72 h. But none of the 72 h clones shows GFPDbox expression (Supplementary Figure 1e). In contrast, the progeny of dronc⁻ cells maintain the GFPDbox label (Supplementary Fig. 1g). These results are included in the revised version (pages 4 and 5)

(2) A very recently published study (Khan et al., PLoS Genetics. July 2017) has demonstrated that, following a damage event in the wing disc, JNK signaling is activated and creates a positive feedback loop through upregulation of the DUOX maturation factor, moladietz. This maintains JNK activity throughout regeneration, days after the damage occurred. These observations unfortunately undermine the novelty of some observations in the current manuscript (permanent JNK activation following transient injury), though the current study does demonstrate that simply activating JNK without cellular/tissue damage is sufficient to maintain its activity. While this is an important new observation, the other work should be cited and this work put into that context

The paper by Khan et al mentioned by the referee is indeed very relevant to our work – we thank the referee for pointing out this paper, which we had not seen when we submitted the original manuscript.

In the revised version we have incorporated the results of experiments to analyse the roles of ROS and mol in the persistence of JNK expression after irradiation of apoptosis-defective cells. The model we present to explain the sustenance of JNK is essentially based on the model Khal et al propose. An important difference is that in our case there is no initial damage and that we deal with apoptosis deficient cells

(3) The current story is somewhat minimal. .. these observations are fundamentally an extension of known relationships: damage activates JNK, p53 can activate JNK, JNK activates JAK-STAT, Wg, and Dpp—the novelty is the extended time over which the response is maintained if apoptosis is blocked.

While we appreciate the technical quality of the referee's report, we disagree strongly with the description that our story is "somewhat minimal". We show that in apoptosis-defective cells a brief JNK activation event (X-radiation. or p53 or HepCA pulses) will result in sustained JNK function that causes tissue overgrowths. To our knowledge this is entirely new.

Moreover, we show that, at least in Drosophila, the overgrowths associated with overexpression of the Ras pathway (known to be involved in many

vertebrate tumours) are due to the same phenomenon of JNK sustenance after stress

In addition, in the revised version we provide new data suggesting a plausible model of JNK sustenance. In this we have benefited greatly from the results and ideas of the recent paper by Khan et al 2017

The first three of five figures all make a similar point—transient cellular insult/activation of JNK signaling leads to prolonged activation of JNK and its downstream effector pathways.

In the revised version we have merged Fig 1 and 2.

(4) The authors do not speculate on the mechanism by which JNK activity might remain high days after the initial stimulus. It seems worth at least mentioning potential mechanisms, and for this level of journal, testing one or a few. For example, the recent Khan et al., (2017) study demonstrated that, following damage to the wing disc, JNK signaling is activated and creates a positive feedback loop through upregulation of the DUOX maturation factor, moladietz. This appears to maintain JNK activity throughout regeneration, days after the damage occurred. It would be interesting to test if moladietz and ROS are involved in the permanent JNK activation following the various treatments presented in the current study. In damaged eye discs, macrophages are recruited to the disc and can activate JNK signaling in the eye disc cells (Fogarty et al., Science Rep. 2013). Could a similar mechanism be at play here? Are macrophages recruited to wing discs following transient JNK activation? Since JNK is known to regulate expression of many metabolic genes, maybe the cells are now stressed from having their metabolism perturbed, which in turn keeps JNK on. Obviously there are numerous possibilities, not all of which would need to be followed, but some attempt to define a mechanism would significantly increase the impact of this work

See our response to point 2 of this referee

We looked for the presence of macrophages in *dronc*⁺ and *dronc*⁻ discs 72 and 96 h after irradiation but could not make a clear conclusion about this

*(5) The experimental design does not seem to allow one to distinguish between the role of initial JNK signaling and prolonged JNK signaling, since the expression of *bskDN* or *puc* occurs throughout, and even precedes, the transient injury. It is known that JNK signaling is required for overgrowth following injury in apoptosis deficient cells (Ryoo et al., Dev Cell. 2004). Experiments need to be designed that allow for the initial JNK activation, but then turn JNK off after some later time point to clearly test the role of the late JNK activity. The controls used likely do not accomplish this, as in a wildtype background, cells with initial high levels of JNK activation die, and when the experimenters do block both apoptosis and JNK activity using *BskDN*, they do it from the start of the experiment. Without this, the major novel contention of the*

paper, that prolonged activity of the JNK pathway (and its downstream effectors) is essential for tissue overgrowth, is not supported by the current data.

This is a reiteration of point 1 by the referee, see our response above.

(6) The current experiments do not definitively show that permanent activation of JNK (in apoptosis deficient cells) is sufficient to induce tissue overgrowth, because they do not quantitate the effect on growth of transiently activating JNK through misexpression of p53 or hep^{CA} (Figure 2). Quantifying overgrowth in these experiments, using the posterior/anterior ratio estimates used in Figs 4+5, would do this

The pictures in Fig. 1 h-k clearly illustrate the effect on growth of JNK induction by p53 and by hep^{CA}. Nearly all the discs of the corresponding genotypes show similar outgrowths and since the relevant factor is JNK activity we did not think it was necessary to present a detailed quantification. Nevertheless, to satisfy the referee we have quantified the results with hep^{CA}: the comparison of the size of posterior compartment in control UAS-GFP tubgal80^{TS}/ UAS- hep^{CA}, hh-Gal4 dronci²⁹/+ and experimental UAS-GFP tubgal80^{TS}/ UAS- hep^{CA}, hh-Gal4 dronci²⁹/dronci²⁴ discs indicates a P/A ratio of 0,37 in controls (n=34) and of 0.43 (n=11) in experimental. The difference is significant (Mann Whitney test, p<0.0001).

Minor points:

Figure 1 – Needs separate channels, at least for H-J. Also, it should be made clear in the Figure, not just the legend, that H-J are irradiated discs 96h. There should at least be a control (no irradiation) for these experiments, and also perhaps a 24hr timepoint.

The pictures the reviewer refers to are now in Supplementary Fig. 1a,c. To satisfy the referee we now show separate channels. Regarding a control discs, we present non-irradiated discs in Fig.1 b,c, that show no JNK activity

Panel H. Minute+ dronc- clones are generated in a M+/- background, however, it is never discussed why the Minute mutation is present, or what consequences that may have on the experiment. Given the role of Minute mutants in cell competition, and the role of JNK signaling in cell competition, it seems that this may affect the results of these experiments. Is the Minute required to generate dronc- clones? Since it is possible to generate homozygous null mutant wing discs (as in Panels C,E,G), why not simply use that approach and include wildtype control discs?

Here we have used the Minute technique (Morata and Ripoll *Develop. Biol.* 42, 1975) to provide dronc mutant clones (that are Minute⁺) with

proliferation advantage, so that they fill the entire posterior compartment. The whole compartment becomes *dronc*⁻ and *Minute*⁺ from early in the development of the disc. Here the control is the anterior compartment of the same disc, which remains *dronc*⁺ and *M*/⁺. There is no cell competition across compartment borders (Simpson and Morata *Develop. Biol.* 85, 1981).

We have successfully used this method in previous publications (Martin and Morata, *Development*, 133, 2006; Ballesteros-Arias et al 2014, cited in the manuscript).

Figure 2 – For both the p53 and hepCA expression experiments, there needs to be an earlier time point showing that JNK was activated in the Dronc+ control discs and goes away by 96hrs.

We have performed an experiment to demonstrate that hepCA does indeed activate JNK in *dronc*⁺ discs. hepCA was active 16 h in the posterior compartment and the discs fixed 2 h later. The result is illustrated in Supplementary Fig 1h. The posterior compartment shows JNK activity.

Figure 3 – Panel C, the expression pattern of nubbin-Gal4 needs to be delineated for readers not familiar with the wing disc and this particular driver (at least provide an outline on the image).

OK, done

Why do panels F,G switch to different colour palette? Should maintain red and green for consistency.

OK, done

Supplemental Fig 2 – In panels A and B, shouldn't the non-GFP area be outlined in yellow, to indicate the region of Gal4 expression where ras is misexpressed? Otherwise it appears that there is more cell death in the region of interest.

OK, done

Discussion. It might be worth speculating about whether JNK can be activated at sub-lethal doses. If not, this may suggest that there is some threshold, beyond which JNK cannot be turned off.

This is an interesting point. Indeed we find that JNK can be activated at sub-lethal doses. It is clearly illustrated in in Supplementary Fig. 1 where we show that cells that have acquired puc (JNK) activity after X-Rays leave progeny that contribute to reconstruct the disc. Since it s not central to the main message of the paper we have not discussed it.

Pg. 4. "This result suggests that once activated, JNK is able to sustain its own expression." Should be "activity" not "expression".

OK, changed

Pg. 5. Why not show the data that JNK is permanently activated following irradiation of p53 mutant discs?

We compared JNK expression in *dronc⁻ p53⁺* and *dronc⁻ p53⁻* disc after X-Rays and did not find a difference. The result is difficult to visualize, as the discs look identical. In the revised version we now mention that irradiated *dronc⁻ p53⁺* and *dronc⁻ p53⁻* discs present similar JNK levels

Pg. 6. "In this experiment, we introduced a lineage tracing cassette (see Methods) to follow the behaviour of rasV12 -expressing cells, which invade the surrounding tissue." The use of "invade" here seems overreaching since it can imply migration, however, these cells may instead result from increased proliferation and subsequent loss of sal-driven GFP. Even in the control (Suppl. Fig 2E), there appear to be sizeable regions of lineage+ cells that do not (or no longer) express sal-driven GFP.

We have removed the word "invade" for "extend"

REVIEWERS' COMMENTS:

Reviewer #1 (Remarks to the Author):

The authors have undertaken a number of new experimental approaches in response to the reviewers' concerns, significantly strengthening the manuscript. Specifically, the use of additional methods to assess JNK signaling, the lineage tracing experiments, and the examination of the roles of ROS and moladietz are strengthen the authors' conclusions.

A note on the authors' response to referee 2: the authors are correct that RHGmiRNA and H99 do not generate undead cells.

Remaining minor concerns:

1. While the manuscript is very well written, some light copyediting would correct errors in the Summary and elsewhere.
2. Page 6 line 159 – does the journal policy allow data not shown?
3. Figure panels 5 a, d, and e do not appear to be called out in the text
4. Throughout the text, figure legends, and figures, degrees should be noted by a degree symbol (°) rather than a superscripted 0 (0)
5. Figure 2f, the Mmp1 panel is unconvincing. Can the area of Mmp1 expression in these discs be quantified? Also, a control panel should be included that shows Mmp1 in dronc- after X ray without molRNAi.
6. Figure 3 legend. Change “permanent” to “sustained” or other more precise term as has been done in the rest of the manuscript.

Reviewer #2 (Remarks to the Author):

The authors have added some new experimental data and addressed most of the issues I raised. I am now satisfied with the revisions and support publication.

Reviewer #3 (Remarks to the Author):

The authors have provided significant improvements to the manuscript, particularly through the many new experiments they performed. The results of these experiments provide additional support for their major contentions, and greatly improve the mechanistic insight of the paper. Therefore, we support publication of the manuscript. We would like to suggest a few minor revisions to the text for the authors to consider:

- a) The results may read more clearly if presented in past tense (e.g., "We compared..." or "We then compared...") instead of the past perfect tense the authors frequently use (e.g., "We have compared...").
- b) In Fig 2c and f, the authors report suppression of prolonged JNK activity by reducing ROS via expression of SOD:Cat or mol RNAi. The authors conclude that these conditions "result in both cases in strong reduction of JNK activity 96 h after the irradiation." However, the images, particularly Fig2f, are not entirely convincing. Perhaps a better image could be used that more

clearly demonstrates the described phenotype. Also, because there is no quantification provided, it is difficult to evaluate if this is really a "strong reduction", especially since there are clearly numerous JNK-active cells still present in the regions of ROS reduction. It might be more accurate to indicate that there is "some reduction". Of course, more compelling images might better support a conclusion of "strong reduction".

c) To assist the community, it would be very helpful if the authors could include more information on reagents used in the study, including stock numbers for fly lines, clone/catalog id for antibodies, etc.

Our response to the referees reports

Reviewer 1

This person considers that the new experiments significantly strengthen the manuscript:

The authors have undertaken a number of new experimental approaches in response to the reviewers' concerns, significantly strengthening the manuscript. Specifically, the use of additional methods to assess JNK signaling, the lineage tracing experiments, and the examination of the roles of ROS and moladietz are strengthen the authors' conclusions.

Remaining minor concerns:

Page 6 line 159 – does the journal policy allow data not shown?

Since it was not essential, we have eliminated the sentence with the “data not shown” words

Figure panels 5 a, d, and e do not appear to be called out in the text

we have modified the text and Fig 5 a, d and e are now mentioned

Throughout the text, figure legends, and figures, degrees should be noted by a degree symbol (°) rather than a superscripted 0 (0)

OK, done

5. Figure 2f, the Mmp1 panel is unconvincing. Can the area of Mmp1 expression in these discs be quantified? Also, a control panel should be included that shows Mmp1 in dronc- after X ray without molRNAi.

We agree that Fig 2f is not totally convincing (reviewer 3 raises a similar point about this panel). We have re-examined all the discs from this experiment (total of 7) and we observe in all of them a diminution of Mmp1 label but not complete suppression, probably because the RNAi used is not totally effective. We have modified the wording in the manuscript to weaken the statement from “very strong reduction” to “reduced JNK activity”

Figure 3 legend. Change “permanent” to “sustained” or other more precise term as has been done in the rest of the manuscript

OK

Reviewer 2

This person is satisfied with the new experimental data and supports publication

The authors have added some new experimental data and addressed most of the issues I raised. I am now satisfied with the revisions and support publication.

Reviewer 3

This person considers that the new experiments included in the revised version greatly improve the mechanistic insight of the paper

The authors have provided significant improvements to the manuscript, particularly through the many new experiments they performed. The results of these experiments provide additional support for their major contentions, and greatly improve the mechanistic insight of the paper. Therefore, we support publication of the manuscript.

Some minor revisions for the authors to consider:

a) *The results may read more clearly if presented in past tense (e.g., "We compared..." or "We then compared...") instead of the past perfect tense the authors frequently use (e.g., "We have compared....").*

OK with thanks

b) *In Fig 2c and f, the authors report suppression of prolonged JNK activity by reducing ROS via expression of SOD:Cat or mol RNAi. The authors conclude that these conditions "result in both cases in strong reduction of JNK activity 96 h after the irradiation." However, the images, particularly Fig2f, are not entirely convincing. Perhaps a better image could be used that more clearly demonstrates the described phenotype. Also, because there is no quantification provided, it is difficult to evaluate if this is really a "strong reduction", especially since there are clearly numerous JNK-active cells still present in the regions of ROS reduction. It might be more accurate to indicate that there is "some reduction". Of course, more compelling images might better support a conclusion of "strong reduction".*

See our response to reviewer 1. We have modified the text (page 6 bottom) and now say "result in both cases in reduced JNK activity 96 h after the irradiation"

c) *To assist the community, it would be very helpful if the authors could include more information on reagents used in the study, including stock numbers for fly lines, clone/catalog id for antibodies, etc.*

Much of this information is included in the Methods of the revised version of the manuscript